

# Using Probability Density Functions to Evaluate Models (PDFEM, v1.0) to compare a biogeochemical model with satellite derived chlorophyll

Bror F. Jönsson[1], Christopher L. Follett[2], Jacob Bien[3], Stephanie Dutkiewicz[2], Sangwon Hyun[3], Gemma Kulk[1], Gael L. Forget[2], Christian Müller[4], Marie-Fanny Racault[1,6], Christopher N. Hill[2], Thomas Jackson[1], and Shubha Sathyendranath[1,5]

[1]Plymouth Marine Laboratory, Prospect Place, Plymouth, PL1 3DH, United Kingdom
[2]Department of Earth, Atmospheric and Planetary Sciences, Massachusetts Institute of Technology, Cambridge, Massachusetts, USA.
[3]Data Sciences and Operations, University of Southern California, Los Angeles, California, USA.
[4]LMU/HMGU Munich, Flatiron Institute, New York
[5]National Centre for Earth Observation, Plymouth Marine Laboratory, Plymouth, PL1 3DH, United Kingdom
[6]School of Environmental Sciences, University of East Anglia, NR4 7TJ, United Kingdom

**Correspondence:** Bror Jönsson (brj@pml.ac.uk)

**Abstract.** Global biogeochemical ocean models are invaluable tools to examine how physical, chemical, and biological processes interact in the ocean. Satellite-derived ocean-color properties, on the other hand, provide observations of the surface ocean with unprecedented coverage and resolution. Advances in our understanding of marine ecosystems and biogeochemistry are strengthened by the combined use of these resources, together with sparse *in situ* data. Recent modeling advances allow sim-
ulation of the spectral properties of phytoplankton and remote-sensing reflectances, bringing model outputs closer to the kind of data that ocean-color satellites can provide. However, comparisons between model outputs and analogous satellite products (e.g. chlorophyll-a) remain problematic: Most evaluations are based on point-by-point comparisons in space and time where spuriously large errors can occur from small spatial and temporal mismatches, whereas global statistics provide no information on how well a model resolves processes at regional scales. Here, we employ a unique suite of methodologies, Probability Den-
sity Functions to Evaluate Models (PDFEM), which generate a robust comparison of these resources. The probability density functions of physical and biological properties of Longhurst's provinces are compared, to evaluate how well a model resolves related processes. Differences in the distributions of chlorophyll-a concentration [mg m$^{-3}$] provide information on matches and mismatches between models and observations. In particular, mismatches help isolate regional sources of discrepancy, which can lead to improving both simulations and satellite algorithms. Furthermore, the use of radiative transfer in the model
to mimic remotely-sensed products facilitate model-observation comparisons of optical properties of the ocean.

## 1   Introduction

Ocean General Circulation Models (OGCMs) with the added ability to simulate biogeochemical and optical processes are providing remarkable opportunities to assess relationships between physical, chemical, biological and optical oceanographic processes, and to identify feedbacks between the Earth's oceans and climate (Doney et al., 2001; Edwards, 2011; Séférian et al.,





2020). These models describe pathways linking biological and chemical standing stocks (state variables) by either resolving physical, chemical, and biological processes explicitly or by parameterizing the fluxes. Current three-dimensional climate-class coupled physical-biological OCGMs have a horizontal resolution that range from 2-3° down to ∼10 km for the global domain, while regional models can resolve horizontal scales down to a few meters. The associated simulated virtual ecosystems have varying complexity from one phytoplankton type to hundreds of different categories of organisms.

One major challenge in the field of ecosystem and biogeochemical modeling is to devise appropriate methods to compare model results with different kinds of observations, especially since it is often not clear whether the comparisons are necessarily of like-for-like quantities (Dutkiewicz et al., 2020a). Mismatches are spread over different temporal and spatial scales, with time lags and spatial shifts that can generate large errors that can be misleading (Doney et al., 2009). Current methods to compare model output with gridded observations such as satellite derived data are normally implemented on point-by-point
match-ups in space and time, which can put an unreasonable penalty on the model due to small local temporal and spatial shifts. A simple but useful statistical metric is the root-mean-square difference (RMSD) summed over all match-ups. This approach can be extended by re-scaling the match-up uncertainty to the data uncertainty and form a cost function (Forget and Wunsch, 2007; Forget and Ponte, 2015), or be explored in frequency space (Forget and Ponte, 2015). The method can be extended to address temporal lags by calculating the deviations between specific time intervals (day, month, season, year) (Doney et al.,
35  2009).

A common method to collate different categories of errors when comparing model-observation match-ups over space and time is the Taylor diagram (Taylor, 2001). Data points in the Taylor space represent correlation coefficient and scaled RMSD as a single point in the first quadrant of a radial plot. The skills of different models or in different regions, time-spans, or variables can be compared and contrasted by presenting each comparison as individual data points on the same diagram (Taylor, 2001).
The concept of showing different statistical metrics as a normalized and unifying figure has been further expanded with more advanced visualizations such as, for example, target diagrams (Jolliff et al., 2009). These methods are quite useful and their concepts can be extended for application in formal data assimilation (Stow et al., 2009). However, all of these techniques can provide spuriously large errors due to potentially small mismatches in time and space between the model and observations. Small process errors, for example, can be magnified by small spatial shifts in locations where spatial gradients are large. Small
shifts in time can equally lead to large errors in RMSD.

Here we present a complementary approach to evaluate OGCMs, where the statistical properties of probability density estimates are used instead of model-observation match-ups. The model probability for finding, for example, chlorophyll-a (Chl) concentration, within a particular interval (in this case 7 years) either globally or in a specific ecological regions is compared with the corresponding probability in the observations, without considering the exact time and location where the
values were found.

The study is formally based on probability density estimates and we use the commonly-used term Probability Density Function (PDF) without assuming that the distribution of values follow any particular statistical probability distribution. An example of this approach is the study by Jönsson et al. (2015) where Net Community Production derived from *in situ* observations were shown to compare well with two biogeochemical models, when zonal ranges of values were compared, while direct match-ups



suggested no skill in the models to reproduce observations. Another example is the work of Mora et al. (2015) where distributions of emergent properties such as phytoplankton community structure and Carbon-to-chlorophyll ratios in the European Regional Seas Ecosystem Model (ERSEM) were compared with the equivalent satellite-derived or in situ properties. Mechanistic insight can be attained by comparing the moments of the probability distributions in different regions (Cael et al., 2018). The suite of methods and accompanying code package is called Probability Density Functions to Evaluate Models (PDFEM)

version 1.0

The rationale for our approach is that the distribution of a certain property has the ability to provide insight into how well a model resolves physical and biological processes, without being penalized for small and often unimportant offsets in space or time. Comparing distributions of different properties simulated by models with corresponding distributions of observations has the potential to illuminate why observations and models diverge. The difference in the shapes of two distributions could

provide clues into how well processes are represented in the models. An absence of long tails in the model-derived distribution when they are seen in observations can, for example, suggest that potentially important but rare events are missing (Jönsson and Salisbury, 2016). Bi-modality in the model-derived distributions, when they are not seen in observations, may indicate that the model solution has unrealistic local equilibria; and the opposite might suggest that processes or water masses that are important in the real world are not resolved by the model. And two similar but shifted distributions might suggest parameterization

problems within the ecosystem model.

To further counteract spurious mismatches from small spatial displacements in point-by-point comparisons, we aggregate data within ecological provinces and compare the statistics of model and observational distributions within each province. This feature-based comparison can be done at many different scales ranging from eddies and fronts to the global scale (Vichi et al., 2011) and has been used in the past to look at, for example, phenology shifts in the North Atlantic Ocean (Henson et al.,

2009). The regions over which to aggregate data could in principle be determined separately for models and observations, but this is challenging in practice since such dynamically defined domains change in time (Reygondeau et al., 2013). Instead, we use static regions that can readily help us to identify processes that are potentially misrepresented in the models by isolating provinces that are expected to be controlled by similar processes and respond to similar sets of physical forcings.

Finally, we need to formulate metrics to compare models with observations. This is particularly challenging when comparing

satellite-derived and model-simulated Chl, since the former is defined as a depth-integrated property dependent on the light attenuation in the water column, which is exponentially weighted towards the surface values (Gordon, 1980; Sathyendranath and Platt, 1989), and model Chl is depth-resolved. The physical-biogeochemical model we use here (Forget et al., 2015; Forget and Ferreira, 2019; Dutkiewicz et al., 2015, 2021; Follett et al., 2021) addresses this challenge by explicitly resolving the light field vertically and simulating Remote Sensing Reflectances ($R_{rs}$, Bailey and Werdell, 2006). The resulting 2D $R_{rs}$

fields are converted to Chl using standard algorithms for satellite-derived Chl (see the methods section). These Chl estimates have analogous benefits and constraints to satellite-derived Chl when it comes to depth integration, taking into account light attenuation in the ocean. The model, in effect, has a simulated satellite field which allows us to readily compare Chl estimates from satellites and models.





The paper is organized with a methods section describing all data and assumptions used in the study followed by an analysis

of 1) The global probability distributions of Chl from the Darwin-CBIOMES-0 configuration, the Ocean Colour – Climate Change Initiative (OC–CCI) satellite-derived Chl product, and *in situ* observations; 2) The PDFs in all non-coastal Longhurst provinces; and 3) The monthly distributions of Chl in four representative Longhurst provinces in the North Atlantic Ocean. Earth Mover's Distance (EMD) is used to quantify the differences in distributions. We end with an overarching discussion of the use of density distributions, Longhurst provinces, and model Chl derived from simulated $R_{rs}$ to assess the skill of

biogeochemical global-ocean models.

## 2 Methods

### 2.1 Biogeochemical Provinces

The analysis is performed after partitioning the global ocean according to the Longhurst (2007) geographical classification system of biomes and provinces. This classification is based on physical and chemical conditions and processes that shape

marine ecosystems over large scales. The Longhurst province system uses a two-level approach with a higher level distinguishing the Coastal biome from the open-ocean biomes, i.e. the Trades, Westerlies, and Polar biomes. The lower level divides each of the coastal and oceanic biomes into provinces that are characterized by similar traits from oceanographic, ecological and topographical perspectives (Longhurst, 2007). The resulting classification, as seen in Figure 1, has 57 distinct biogeochemical provinces (BGCPs) with generally high internal homogeneity and high external heterogeneity in marine biodiversity

(Longhurst, 2007; Beaugrand et al., 2000; Reygondeau et al., 2013). The original Longhurst provinces are static in time and space and the definition of the province boundaries included qualitative criteria. It is, therefore, possible that the boundaries between provinces could be located differently if objective, quantitative criteria were used. The dynamic nature of the boundaries is not explored in the current study, but is an important area for future research. Our study uses Longhurst provinces as the basis of the comparisons, assuming that they provide a reasonable partitioning of regions with similar physical and ecological

characteristics.

[Figure 1 about here.]

### 2.2 The Darwin-CBIOMES-0 physical-biogeochemical-optical model

We use output from a coupled physical-biogeochemical-optical model adapted and configured for the Simons Collaboration on Computational Biogeochemical Modeling of Marine Ecosystems (CBIOMES) project. The model configuration, hereafter

denoted Darwin-CBIOMES-0 (Dutkiewicz, 2018), is global and simulates the period 1992-2006 (Forget and Ferreira, 2019). The physical component uses the MITgcm (Campin et al., 2020) in a 3-dimensional global configuration developed as part of the Estimating the Circulation and Climate of the Ocean project (Forget et al., 2015; Forget and Ponte, 2015, ECCOv4). The state estimate uses a "least-squares with Lagrangian multipliers" approach to adjust internal model parameters, as well as initial and boundary conditions with global observational data streams, including satellite altimetry and Argo floats. The resolution



is nominally 1 degree in the horizontal, and ranges from 10 m in the vertical at the surface to 500 m at depth (see Forget et al., 2015, for details).

The biogeochemical component resolves the cycling of carbon, phosphorus, nitrogen, silica, iron, and oxygen through inorganic, living, dissolved and particulate organic phases. The ecosystem incorporates 35 phytoplankton and 16 zooplankton types as in Dutkiewicz et al. (2021). The phytoplankton include several biogeochemical functional groups: Diatoms (that utilize

silicic acid), coccolithophores (that calcify), mixotrophs (that photosynthesize and graze on other plankton), nitrogen fixing cyanobacteria (diazotrophs), and pico-phytoplankton. Each group has a range of size classes such that the phytoplankton span from 0.6 to 228 $\mu$m equivalent spherical diameter (ESD). Several phytoplankton parameters, including maximum growth rate, nutrient affinity, and sinking are expressed as functions of cell volume, though with distinct differences between functional groups as suggested by observations (Dutkiewicz et al., 2020b; Sommer et al., 2017). The 16 size classes of zooplankton range

from 6.6 $\mu$m to 2425 $\mu$m ESD, and graze on (phyto- or zoo-)plankton 5 to 20 times smaller than themselves, but preferentially 10 times smaller (Hansen et al., 1997; Kiørboe, 2019; Schartau et al., 2010) with a Holling III parameterization (Holling, 1959). The simulation uses Monod kinetics, and C:N:P:Fe stoichiometries are constant over time (though differ between phytoplankton groups). We refer the reader to Dutkiewicz et al. (2015, 2020b, 2021) for a further description of the model, as well as evaluation of the modelled plankton size class and functional group distributions. Here we focus instead on the model Chl con-

centrations. Each of the 35 phytoplankton types have dynamic Chl that alters as a function of light, nutrients and temperature following Geider et al. (1998). Chl$_{mod}$ refers to the sum of modelled Chl-a across all the phytoplankton types.

The optical component of the model includes explicit radiative transfer of spectral irradiance in 25 nm bands between 400 and 700 nm. The three-stream (downward direct, $E_d$, downward diffuse, $E_s$, and upwelling, $E_u$) model (following Aas, 1987; Gregg, 2002; Gregg and Casey, 2009) is reduced to a tri-diagonal system that is solved explicitly (see Dutkiewicz et al. (2015)

for more details). In-water irradiance fields are altered by the spectral absorption and scattering by water molecules, the 35 phytoplankton types, detritus and coloured dissolved organic matter (CDOM). Irradiance just below the surface of the ocean (direct, $E_{do}$, and diffuse, $E_{so}$, downward) is provided by the Ocean-Atmosphere Spectral Irradiance Model (Gregg, 2002; Gregg and Casey, 2009, OASIM).

Output from the optical model include spectral surface upwelling irradiance similar to that measured by ocean color satellites

(Dutkiewicz et al., 2018, 2019). As in these earlier studies, we calculate model reflectance for each waveband as the upwelling just below the surface ($E_u$) divided by the total downward (direct and diffuse) irradiance also just below the surface (as provided by OASIM): $R(\lambda, 0^-) = \frac{(E_u(\lambda))}{(E_{d0}(\lambda) + E_{s0}(\lambda))}$. We first convert model subsurface irradiance reflectance to remotely-sensed reflectance just below the surface using a bidirectional function Q: $R_{rs}(\lambda, 0^-) = \frac{R(\lambda, 0^-)}{Q}$, where we assume that Q = 3 sr (as in Dutkiewicz et al., 2019; Gregg, 2002). Secondly we convert $R_{rs}(\lambda, 0^-)$ to above-surface remotely-sensed reflectance

$R_{rs}(\lambda, 0^+)$ using the formula of Lee et al. (2002): $R_{rs}(\lambda, 0^+) = \frac{0.53 R_{rs}(\lambda, 0^-)}{((1 - 1.7 R_{rs}(\lambda, 0^-)))}$. These spectral fields will be referred to as model $R_{rs}$ (units of sr$^{-1}$) and is comparable to the $R_{rs}$ provided by ocean color satellite databases.

An advantage of having the model $R_{rs}$ is that we can provide a model-derived satellite-like Chl similar to that described in the previous section (see Dutkiewicz et al., 2018, 2019). In practice we interpolate the 13 wavebands of $R_{rs}$ from the model to the same bands as used in OC–CCI, and use the maximum ratio of the blue signal to the green and a 4th order polynomial to





estimate the satellite-like derived Chl (following O'Reilly et al., 1998). Here, for simplicity, we use the OC version 2 algorithm and coefficients. This product is termed $\text{Chl}_{Rrs}$ in this paper and is technically more comparable to the real-world satellite product than the model "actual" Chl at the surface ($\text{Chl}_{mod}$, see further discussion in Dutkiewicz et al., 2018). Any pixels with invalid data in the satellite product after downscaling to the model grid is masked in the corresponding model output.

## 2.3 Satellite-derived Chlorophyll

The model is compared to satellite-derived Chl products originally at 4 km resolution from version 4.2 of the Ocean Colour Climate Change Initiative (Mélin et al., 2017; Sathyendranath et al., 2019, 2020, OC–CCI). This is a blended Chl product where data from the Sea-viewing Wide-Field-of-view Sensor (SeaWiFS), the Aqua MOderate-resolution Imaging Spectro-radiometer (MODIS-Aqua), the MEdium spectral Resolution Imaging Spectrometer (MERIS), and the Suomo-NPP Visible Infrared Imaging Radiometer Suite (NPP-VIIRS) are merged into a unified product. SeaWiFS operated from September 1997

until December 2010 and MERIS from March 2002 to May 2012, MODIS-Aqua was launched in May 2002 and VIIRS in October 2011; the latter two sensors are still operational as of December 2021. Data from the different instruments are merged after band-shifting normalized remote-sensing reflectance ($R_{rs}$) to the spectral bands of SeaWIFS and correcting for inter-sensor biases. Atmospheric correction is performed using POLYMER v3.5 (Steinmetz et al., 2011) for MERIS and MODIS-A, and NASA/L2Gen 7.3 for SeaWiFS and VIIRS. All individual grid cells are classified optically using a fuzzy-logic approach

(Moore et al., 2009, 2012; Jackson et al., 2017) and a combination of the best Chl algorithms for each class is used along with class membership at each pixel to generate Chl at each pixel. The spatial mapping follows NASA protocol for level 3 processing by considering a 4-km bin as valid if there is at least a single 1-km valid pixel in that bin from at least one sensor, and taking the mean if more than one valid observation is available. The resulting time series for the period between 1997 and 2019 is designed to be internally consistent (all radiometric products band-shifted to a common set of bands corresponding to

SeaWiFS) and stable (corrected for inter-sensor bias Sathyendranath et al., 2019). The resulting daily 4-km OC-CCI product is downscaled to the Darwin-CBIOMES-0 grid using bucket resampling from the SatPy resampling package (Raspaud et al., 2019) in Python. We denote the Chl product from OC-CCI $\text{Chl}_{sat}$ henceforth in the study.

## 2.4 *In situ* data

The satellite-derived and simulated Chl concentrations are matched with ∼80,000 *in situ* Chl observations for comparison.

The *in situ* Chl data set is based on a global compilation developed to evaluate the quality of ocean color satellite data records and to evaluate ocean color products from OC–CCI (Valente et al., 2019a). The observations were acquired from several sources including MOBY, BOUSSOLE, AERONET-OC, SeaBASS, NOMAD, MERMAID, AMT, ICES, HOT, GeP&CO, AWI, ARCSSPP, BARENTSSEA, BATS, BIOCHEM, BODC, CALCOFI, CCELTER, CIMT, COASTCOLOUR, ESTOC, IMOS, MAREDAT, PALMER, SEADATANET, TPSS, and TARA. The data set spans the period from 1997 to 2018 and vari-

ables include spectral remote-sensing reflectances, concentrations of Chl, spectral inherent optical properties, spectral diffuse attenuation coefficient, and total suspended matter. Different methodologies have been implemented for homogenization, quality control and merging of all data. Observations close in time and space are averaged and some data were eliminated after





failing quality control. To be consistent with satellite-derived Chl values, which are derived from the light emerging from the upper layer of the ocean, all observations in the top 10 m (replicates at the same depth, or measurements at multiple depths) were averaged. Data points are discarded if the coefficient of variation among observations is more than 50%. The compiled *in situ* data set is publicly available (Valente et al., 2019b). The resulting data product is referenced to here as $\text{Chl}_{obs}$.

## 2.5 Statistical analysis

While the objective of this study is to assess different continuous probability distributions of Chl, we perform all statistical analyses by converting the distributions to discrete histograms where the data are divided into equally sized bins on a log-scale. The reason for using a log-scale is that Chl concentrations can be assumed to follow a log-normal distribution at a variety of spatial and temporal scales (Campbell, 1995) and this transformation allows for better characterizations at low concentrations. We use the same set of 100 equally-sized bins from $-6.9$ ($\ln 0.001$) to $4.61$ ($\ln 100$) for all data sources and in all calculations. The histograms are generated by binning daily interpolated Chl from both satellite (derived Chl from OC–CCI) and the Darwin-CBIOMES-0 output by month and Longhurst province for the period 1998 to 2007. The *in situ* data set has very few, if any, observations, in several provinces and is only used for comparisons on global or biome scales. Percentiles, medians, standard deviations (SDs), and other statistics are all calculated from the resulting histograms using specifically-developed code.

### 2.5.1 Earth Mover's Distances

We leverage Earth Mover's Distance (Rubner et al., 2000, EMD) to quantify the difference between different distributions. EMD, also known as the Wasserstein metric in mathematics (Vaserstein, 1969) and Mallow's distance in statistics (Levina and Bickel, 2001), is a popular optimal transport method (Monge, 1781) for measuring the distance between two probability distributions, widely used in image processing (Frogner et al., 2015) and scientific applications (Orlova et al., 2016). The distance is based on imagining a mound of dirt shaped like the first distribution and considering how much effort would be required to transform it to the second distribution's shape. Given a distance metric in this space (in this case the absolute difference in log-Chl), it is possible to calculate the minimum redistribution of mass needed to transform one probability distribution to the other. EMD measures the total sum of such an optimally-planned transfer of mass. Rather than focusing on the distance between any particular aspect of the distributions such as their means or variances, EMD provides a more comprehensive measure of distance. For computation, the original log-Chl measurements in each province are transformed into histograms using the earlier mentioned bin definitions. We use the Python package `pyEMD` (Pele and Werman, 2008, 2009; Mayner et al., 2015) to calculate EMDs between the histograms. All reported EMDs have the natural log of Chl as unit.



# 3 Results

## 3.1 Taylor Diagrams

Taylor diagrams (Taylor, 2001) based on different spatial and temporal aggregations of satellite and model chlorophyll concentrations reveal different model bias patterns at different scales of averaging (Figure 2). These diagrams are polar representations of pairwise statistics between $\mathrm{Chl}_{sat}$, $\mathrm{Chl}_{mod}$, and $\mathrm{Chl}_{Rrs}$ with the correlation coefficient as the angle and the standard deviation (SD) of $\mathrm{Chl}_{mod}$ or $\mathrm{Chl}_{Rrs}$ normalized to $\mathrm{Chl}_{sat}$ as the radius. Each individual datapoint represents the SD and correlation for a specific Longhurst province, different colors denote different basins and different marker shapes denote different biomes. Figure 2, panels A–B shows the resulting diagram based on daily match-ups for individual grid cells in the model versus satellite data re-projected to the model grid. Each symbol shows the point-by-point statistics within a Longhurst province. We find the correlation to generally be quite low (R=0–0.5) for all Longhurst provinces, and model SD to be 0.5 to 1.8 times $\mathrm{Chl}_{sat}$. This skill metric could be highly affected by small mismatches and lags in time and space between the model and satellite data. We utilize the assumption that a given Longhurst province is controlled by a specific combination of physical, chemical, and biological process by averaging $\mathrm{Chl}_{Rrs}$, $\mathrm{Chl}_{sat}$, $\mathrm{Chl}_{sat}$ over all grid cells in each province for each day and present the resulting dataset as Taylor diagram as seen in Figure 2, panels C–D. Some Longhurst provinces show a better correlation (R>0.8), whereas others have a negative correlation between satellite derived Chl and model output. Model SD scaled to satellite data also show more variability between provinces compared with individual grid cells. While aggregating the data over Longhurst provinces dampens random spatial errors, temporal fluctuations at the daily scale are still present. In Figure 2, panels E–F, we show a Taylor diagram using monthly time series instead of the daily time series used for Figure 2, panels C–D and find a much clearer separation between the Longhurst provinces. Some provinces are highly correlated, while others are showing negative correlations. The latter patterns could be explained by the more aggregated data sets having less random noise, which let systematic mismatches to be more visible and detected by the Taylor diagram. We find that $\mathrm{Chl}_{mod}$ generally have a slightly more pronounced spread, and hence more variability in the misfit than $\mathrm{Chl}_{Rrs}$.

[Figure 2 about here.]

## 3.2 Global Distributions

Global distributions of $\mathrm{Chl}_{obs}$, $\mathrm{Chl}_{sat}$, $\mathrm{Chl}_{mod}$, and $\mathrm{Chl}_{Rrs}$ show that distributions of satellite derived Chl and *in situ* observations are very similar, but Darwin-CBIOMES-0-based Chl show a systematic bias (Figure 3). Histograms of $\mathrm{Chl}_{obs}$ and $\mathrm{Chl}_{sat}$ are shown in Figure 3, panels A and B. Only data pairs where both $\mathrm{Chl}_{obs}$ and $\mathrm{Chl}_{sat}$ have valid values are used (35,174 match-ups out of 80,524 observations). We find the *in situ* and satellite datasets to have similar distributions without any significant biases in $\mathrm{Chl}_{sat}$. Note that some of the Chl algorithms contributing to the final $\mathrm{Chl}_{sat}$ product would have been tuned using a small subset of the *in situ* database, and corresponding *in situ* or satellite-derived $R_{rs}$ (other than the OC-CCI products). Similarly, the algorithm used to provide $\mathrm{Chl}_{Rrs}$ from the Darwin-CBIOMES-0 output (O'Reilly et al., 1998) would have used





a subset of the *in situ* dataset along with measured $R_{rs}$ values to calibrate the algorithm. But none of the observations in the *in situ* datasets have been used to tune either the OC-CCI products or the Darwin-CBIOMES-0 outputs.

Whereas $\mathrm{Chl}_{obs}$ closely follows a log-normal distribution, $\mathrm{Chl}_{sat}$ shows some divergence from the expected distribution. The pronounced secondary peak at about 5 mg m$^{-3}$ is related to coastal provinces and the broader peak at 0.5 mg m$^{-3}$ represent

values from the subtropical gyres. In Figure 3B, $\mathrm{Chl}_{sat}$ has thinner tails than $\mathrm{Chl}_{obs}$ and the distribution is more centered around the median. This pattern is consistent with our expectations since $\mathrm{Chl}_{sat}$ has a coarser resolution (4 km) than $\mathrm{Chl}_{obs}$ (the volume of each sample). One can expect a spatially-aggregated measurement to have less extreme values. Panels C and D in Figure 3 are analogous to panels A and B, but with $\mathrm{Chl}_{mod}$ and $\mathrm{Chl}_{Rrs}$ included. Here, collocation between all four sources is required, which results in 20,935 match-ups. The differences in the distributions of $\mathrm{Chl}_{obs}$ and $\mathrm{Chl}_{sat}$ in Figure 3C and

D compared with panels A and B occurs from the shorter time span of the Darwin-CBIOMES-0 configuration (1998-2006) compared with the OC-CCI time domain (1998-2020) and from the masking out of near-coastal locations. The model grid also has a coarser resolution ($\approx 1°$) than the satellite product. The 4-way match-up (Figure 3C and D) allows us to compare the different data sources in a reasonably objective way considering both seasonal variability and data density. We find the global distributions of $\mathrm{Chl}_{mod}$ and $\mathrm{Chl}_{Rrs}$ to be nearly identical to each other, but significantly different from both $\mathrm{Chl}_{obs}$ and $\mathrm{Chl}_{sat}$.

It is clear that Darwin-CBIOMES-0 systematically generates lower Chl concentrations than either the satellite-derived product or *in situ* observations. It is somewhat non-intuitive that the model has similar or longer tails than $\mathrm{Chl}_{obs}$ or $\mathrm{Chl}_{sat}$, considering the model's coarser spatial resolution and the earlier comparison between satellite and *in-situ* Chl. EMDs calculated for $\mathrm{Chl}_{obs}$ vs $\mathrm{Chl}_{sat}$, $\mathrm{Chl}_{Rrs}$, and $\mathrm{Chl}_{mod}$, respectively, (Table 1) confirms these findings with much larger (and similar) distances for $\mathrm{Chl}_{Rrs}$ and $\mathrm{Chl}_{mod}$ than $\mathrm{Chl}_{sat}$. These dissimilar EMDs are the combined result of differences in medians, SD, and skewness

between the model and observations.

[Figure 3 about here.]

While the distributions of $\mathrm{Chl}_{mod}$ and $\mathrm{Chl}_{Rrs}$ are close to identical, there is much more variability between the two properties when compared on a point-by-point basis. Figure 4, panel A, shows a 2D histogram of $\mathrm{Chl}_{mod}$ and $\mathrm{Chl}_{Rrs}$ sampled from the model at the same time and grid cell. While most values fall close to the 1–1, there is a large spread. The 95% confidence

interval of the residual is about two orders of magnitude. These results show the importance of diagnosing the model using a metric that's comparable to $\mathrm{Chl}_{sat}$. Any divergence from the 1–1 line in Figure 4, panel A could incorrectly be interpreted as model misfit.

[Figure 4 about here.]

### 3.3   Distributions in Different Biomes

As the global distributions show only general biases between Darwin-CBIOMES-0, *in situ* and satellite-derived Chl, we divide the data into biomes based on Longhurst (2007) to better understand the misfits. Conclusions about model performance will be different if the shift in the probability distributions is caused by errors which are global, or limited to specific regions of



the ocean. First we compare EMDs calculated using $\mathrm{Chl}_{Rrs}$ and $\mathrm{Chl}_{mod}$ for each month and province, as shown in Figure 4, panel B. We find that Chl distributions from Longhurst provinces in the Westerlies biome are similar, but that provinces in

the Trades biome and, particularly, the Polar biome show larger EMDs between $\mathrm{Chl}_{sat}$ and $\mathrm{Chl}_{mod}$ than $\mathrm{Chl}_{sat}$ and $\mathrm{Chl}_{Rrs}$. Provinces in coastal biome is omitted here and in the further analysis since Darwin-CBIOMES-0 is not developed to resolve coastal processes.

Figure 5 shows cumulative distributions, analogous to Figure 3D, for the Polar (3,741 matches), Westerlies (1,240 matches), Trades (3,328 matches), and Coastal (12,418 matches) biomes. Mismatches are different for the different biomes, with $\mathrm{Chl}_{sat}$

generally following $\mathrm{Chl}_{obs}$ more closely than the model does. The general trend of $\mathrm{Chl}_{sat}$ having less variance than $\mathrm{Chl}_{obs}$ and both $\mathrm{Chl}_{mod}$ and $\mathrm{Chl}_{Rrs}$ having a negative bias is also evident. $\mathrm{Chl}_{mod}$ has a much wider distribution in the Polar biome than $\mathrm{Chl}_{sat}$ or $\mathrm{Chl}_{Rrs}$, especially for low concentrations. All distributions show close similarities in the Westerlies biome, suggesting that the model has a relatively high skill in simulating phytoplankton biomass (for which Chl act as a proxy) in this biome. The largest mismatch between the model and observations is in the Coastal biome, something which is to be expected

considering the spatial resolution of the model. This biome is also where $\mathrm{Chl}_{sat}$ shows the largest inconsistencies compared with $\mathrm{Chl}_{obs}$. Coastal areas tend to have more complex case II waters where Chl algorithms are more affected by Colored Dissolved Organic Matter (CDOM) or Total Suspended Matter (TSM) (Morel and Prieur, 1977; Lee and Hu, 2006).

[Figure 5 about here.]

[Table 1 about here.]

## 3.4 Individual Longhurst provinces

We extend the study to individual Longhurst provinces to further explore differences between the model and satellte-derived observations. The limited geographical and seasonal coverage of *in situ* data limits our ability to include $Chl_{obs}$, we therefore focus on comparing $\mathrm{Chl}_{sat}$ with $\mathrm{Chl}_{mod}$ and $\mathrm{Chl}_{Rrs}$ in the following section while conceding the limitations of the approach: $\mathrm{Chl}_{sat}$, for example, probably underestimates the variance in $\mathrm{Chl}_{obs}$, but is at the same time more representative of the reduced

variance within large model grids. We exclude coastal provinces from the analyses since the model is not expected to have as much skill near land or in shallow waters due to the relatively coarse vertical and horizontal grid resolution earlier discussed, and the challenges with satellite retrieval of Chl in coastal waters. We still find interesting patterns when comparing Chl distributions in individual open-ocean Longhurst provinces.

[Figure 6 about here.]

Individual histograms for Longhurst provinces in the Polar biome often show stark differences between Darwin-CBIOMES-0 and OCCCI, as seen in Figure 6. The very low Chl concentrations seen in the Polar biome (Figure 5) for the $\mathrm{Chl}_{mod}$ distribution primarily occurs in the Boreal Polar, Arctic, and Subarctic sections of the Atlantic Ocean (Figure 6). The Arctic section of the Pacific Ocean shows a small bias towards low values as well, but much less extreme. Chl concentrations are biased low in Darwin-CBIOMES-0 close to the Antarctic continent (Austral Polar province), but biased high towards the Antarctic



Circumpolar Current (Antarctic province). This pattern could suggest a meridional misalignment of physical processes that drive Chl variability in the Antarctic Ocean. Since the comparisons are carried out only for match-up data where satellite and model data are available, the differences observed here cannot be explained as a consequence of poor sampling of the Polar biome by satellites due to adverse viewing conditions, especially in winter (Jönsson et al., 2020b).

[Figure 7 about here.]

Whereas Figure 5 shows a strong correspondence between $Chl_{sat}$ and $Chl_{Rrs}$ in the Westerlies biome, we find a more complicated picture in the individual Longhurst provinces within this biome (Figure 7). In most provinces, $Chl_{sat}$ and $Chl_{Rrs}$ have similar medians, but the tails are often significantly different. Chl concentrations in the Darwin-CBIOMES-0 configuration are biased to high values in most provinces in the Pacific Ocean and the Southern Ocean. $Chl_{Rrs}$ also tends to have a bi-modal distribution in contrast to the expected log-normal distribution of $Chl_{sat}$. There is also a clear negative bias in $Chl_{Rrs}$ in the

Mediterranean Sea, which may indicate that the model resolution is too coarse to resolve all the hydrodynamics in this small, but very complex sea.

[Figure 8 about here.]

  Longhurst provinces in the Trades biome generally show similar distribution widths and shapes of $Chl_{sat}$ and $Chl_{Rrs}$ aside from biases in the medians (Figure 8). The main outliers are the North Tropical Gyre in the Atlantic Ocean and the Archipelagic

Deep Basin in the Pacific Ocean, where $Chl_{Rrs}$ has a rectangular or weakly bi-modal distribution. The Western Warm Pool and South Gyre in the Pacific Ocean also stand out as the only two provinces where $Chl_{sat}$ has a bi-modal distribution and $Chl_{Rrs}$ does not.

### 3.5 Monthly Distributions

  By extending the analysis to monthly distributions for each Longhurst province, we can identify temporal differences between

OC-CCI and Darwin-CBIOMES-0. The large number of resulting probability distributions is challenging to present, and we only show a few interesting examples here. Figures for all provinces are provided with the associated datasets described in the acknowledgements. Figure 9 shows graphical representations of probability distributions for some representative provinces in the North Atlantic Ocean, generated by aggregating daily $Chl_{sat}$, $Chl_{Rrs}$, and $Chl_{mod}$ by climatological month. The largest intra-annual differences are found in provinces in the Polar biome, as exemplified by the Atlantic Arctic province (Figure 9,

panel A), where $Chl_{mod}$ shows much lower concentrations and a much more variability than either $Chl_{sat}$ or $Chl_{Rrs}$ during the winter and spring months. All three data products have similar distributions and progressively smaller variability during the summer. But $Chl_{sat}$ shows both a significant drop in concentrations and an increase in variability in November, not seen in either $Chl_{Rrs}$ or $Chl_{mod}$.

[Figure 9 about here.]

The Gulf Stream province, our chosen example for the Westerlies biome, has a distinct seasonal progression with elevated Chl concentrations during the spring bloom and low values in the summer (Figure 9, panel B). Both $Chl_{Rrs}$ and $Ch_{mod}$





generally compare well with $\text{Chl}_{Rrs}$, but with a negative bias during summer. The model products also tend to have a more stretched out distribution for low values. The good agreement in the seasonal cycle somewhat hides misfits between the model and satellite observations for individual months.

The two provinces representing the Trades biome (the Western Tropical Atlantic province and the Caribbean province) both show a weak seasonal cycle, but very different misfits. The Western Tropical Atlantic province (Figure 9 panel C) has a notable similarity in medians between the different data sources, with the exception that the 99% percentile for $\text{Chl}_{sat}$ is much higher than $\text{Chl}_{Rrs}$ or $\text{Chl}_{mod}$ during spring and summer. This pattern could be interpreted such that processes generating rare bloom events during that time of the year are missing in the Darwin-CBIOMES-0 configuration. The probability distributions of

$\text{Chl}_{sat}$ in the Caribbean province (Figure 9, panel D) are notably different from $\text{Chl}_{Rrs}$ and $\text{Chl}_{mod}$, with both much higher medians and higher 99% percentiles. This pattern is not unique to the province and can be seen for example in the North Atlantic Tropical Gyre province as well (data not shown). The asymmetric shape of the probability distributions of $\text{Chl}_{sat}$ with a high positive skewness is surprising and deserves closer examination in a future study.

## 3.6  Earth Mover's Distance

The calculated EMD distances between $\text{Chl}_{sat}$ and $\text{Chl}_{mod}$ or $\text{Chl}_{mod}$, respectively, for all province-month combinations (partly presented in Figure 9) are shown as maps in Figure 10 for two selected months – January and July – using color intensity to depict the EMD between the two probability density functions for that province-month pair. Here, EMDs provide aggregated information about how different the respective distributions are including mean biases, SD, and skewness. Focusing our attention on the six province-month pairs with highest EMD, we compare the two Chl distributions as overlaid probability

histograms in blue and green (Figure 11). In each of the panels in Figure 11, the two data sources have visibly different Chl distributions, and it is clear that a large amount of probability mass needs to "move" for the two data sources to match. Out of the six, the Atlantic Arctic Province in March is of particular interest with the two distributions of Chl having similar means values, while the EMD is very large. This discrepancy is due to a considerable difference in the second or higher moments between the two distributions – the green distribution ($\text{Chl}_{mod}$) has a much higher spread than the blue ($\text{Chl}_{sat}$). This demonstrates how

EMD can be an effective scalar measure for summarizing the *full* distributional difference between two data sources.

[Figure 10 about here.]

[Figure 11 about here.]

## 4  Discussion

We combined satellite-derived Chl from OC-CCI in combination with *in situ* observations and model output from the Darwin-

CBIOMES-0 configuration to investigate three new approaches for comparing biogeochemical models and observations: The use of Chl proxies analogous to satellite-derived properties instead of directly diagnosed Chl in the model; the utility of Earth Movers Distances (EMDs) as a metric for quantifying differences between distributions; and probability distributions instead



of point-by-point comparisons. The first approach has already been presented and evaluated in Dutkiewicz et al. (2018, 2019) as a general tool, and we will focus on its use in the context of comparing distributions.

The main differences between $Chl_{Rrs}$ and $Chl_{mod}$ occur on regional scales (Figures 3–8), particularly in the Polar and Trades biomes, and these differences generally even out when aggregated over full biomes or globally (Figures 3–5, and Table 1). The differences seen in the Polar biome puts earlier results by Jönsson et al. (2015) – that ecosystem models underestimates winter phytoplankton biomass in the Southern Ocean – in a new light. It is possible that the use of $Chl_{Rrs}$ and $Chl_{sat}$ as a proxy for phytoplankton biomass is biased in these regions by an inability to detect the low concentrations predicted by the models,

something that must be further explored with *in situ* observations. While this issue might seem irrelevant due to the low Chl concentrations, it has large potential effects on the skill in simulating the seasonal progression of Polar ecosystems. Order of magnitude errors in regions with seasonally low Chl concentrations and large annual variability can be critical when the growth is exponential, potentially requiring unrealistically high growth rates or leading to delays in the spring bloom. $Chl_{Rrs}$ is not necessary a more "truthful" diagnostic than $Chl_{mod}$, only closer to $Chl_{sat}$, and quite possibly inherits biases from satellite-

derived proxies. Another motivation to use $Chl_{Rrs}$ as the property of comparison is that $Chl_{mod}$ is poorly defined since the conversion from a depth resolved field to a 2D concentration might be performed differently between different models.

EMDs provide a systematic and quantitative way to assess how the distribution of Chl differs between OC-CCI and the Darwin-CBIOMES-0 configuration. One major application is the ability to compare the likeliness between different distributions in an integrated fashion, as the maps presented in Figure 10. We find that the biggest differences between $Chl_{Rrs}$ and

$Chl_{mod}$ occur in the Polar biome during winter. This pattern is supported by the scatter of EMDs shown in Figure 4, panel B, where provinces in the Polar and Trades biomes tend to have larger EMDs for $Chl_{mod}$ than $Chl_{Rrs}$ when compared to $Chl_{sat}$.

When comparing probability distributions between the Darwin-CBIOMES-0 configuration, OC–CCI satellite-derived Chl product, and *in situ* observations of Chl, we find many similarities, but also important differences. Comparisons between $Chl_{sat}$ and $Chl_{obs}$ (Figures 3 and 5) shows an interesting pattern where OC–CCI diverges from the expected log-normal distribution

with a smaller variance than the set of observations. This difference could be explained by OC–CCI being based on satellite products with overpasses close to noon, limiting the ability to resolve the diurnal cycle, and 4 km-sized pixels that aggregates variability on smaller scales. $Chl_{obs}$ data, on the other hand, are sampled at any time over the day and generally represent a water volume of less than 1 m$^3$. A small water volume has a higher chance to have outliers due to patchiness, while a distribution of satellite-derived Chl observations is more clustered around the population mean from averaging many small

patches over a larger area. This difference is expected by the law of large numbers. It should also be noted that some of the mismatches between Darwin-CBIOMES-0 and observations might be partly explained by the unevenness in temporal and spatial coverage of observations. By only including satellite derived and modeled Chl concentrations, we are able to minimize potential problems with temporal and spatial representativeness of *in situ* observations since $Chl_{sat}$ is interpolated to the grid of Darwin-CBIOMES-0 and only pixels with valid data in both data sources are used.

While the distributions of Chl in a direct match-up between $Chl_{obs}$ and $Chl_{sat}$ vs $Chl_{Rrs}$ and $Chl_{mod}$ suggests that the model underestimates Chl significantly, regional comparisons provides a more nuanced picture. Data from the Westerlies biome have for example almost identical distributions. The largest discrepancies are found in coastal areas and the provinces in the Polar





biome, both of which are notoriously challenging to model due to complex hydrology and large seasonal variability in forcings (light, freshwater run-off from land, nutrient input, etc.). Provinces in the Trades biome generally show less seasonal average

variability, but larger differences in the high and low extreme values. These patterns are evident both in the EMD maps seen in Figure 10 and in the individual histograms seen in Figures 6–8.

Model Chl had long tails with low values in the provinces in the Polar biome, that might be connected to the Darwin-CBIOMES-0 configuration overestimating respiration during winter or possibly exaggerating mixing (cf Jönsson et al., 2015). The tendency of bimodality in PDFs from data generated by Darwin-CBIOMES-0 in the provinces in the Polar biome suggested

that the model shows different distinct states in the phytoplankton community. It is not clear if this pattern is due to the formulation and/or parameterization of the ecosystem model, or due to problems with the high-latitude retrieval of Chl by Polar orbiting satellites at the beginning and end of the growing season (Jönsson et al., 2020a). In any case, our combined use of Longhurst provinces, distributions, and EMD has allowed us to pose differences between models and observations in a way which can be directly analyzed and tested.

The tendency for bi-modal distributions in PDFs generated by the Darwin-CBIOMES-0 configuration also occurs in Longhurst provinces in the Westerlies biome. Here, the differences between satellite-derived and modeled Chl is less clear, with some provinces having extremely similar distributional shapes and others mainly having different variances. The province with biggest differences is the Mediterranean Sea, a result that is not surprising considering the complex hydrology and distinct ecosystem dynamics there. Longhurst provinces in the Trades biome shows generally good fits between the model and OC–

CCI. Provinces in this biome showed a general pattern where model and satellite-derived Chl distributions had similar shapes, but with an offset relative to each other. The biggest differences in the Trades biome are found in the Caribbean and the adjacent North Tropical Gyre. Both provinces shows significantly lower Chl concentrations in Darwin-CBIOMES-0 and tends to be skewed towards low values with longer tails.

Dividing the different datasets into monthly distributions allowed us to further diagnose possible differences between OC–

CCI and the Darwin-CBIOMES-0 configuration. We found that provinces in the Polar biome tended to show the largest discrepancies during winter and spring, a pattern consistent with results by Jönsson et al. (2015). It is also notable that these provinces and seasons were where and when simulated Chl in Darwin-CBIOMES-0 differs the most from $R_{rs}$-derived Chl in the model. These misfits can be due to a number of factors such as inadequate model inputs, forcings, parameterizations, numerical schemes, problems arising from bio-optical constraints due to extreme light conditions, unresolved physical processes,

or a combination of these. Two specific causes suggested by Jönsson et al. (2015) are a meridional misalignment of the physical processes that drive Chl variability in the Antarctic Ocean, or a lack of small-scale variability in the mixed layer dynamics. The latter explanation is supported by comparing distributions of mixed layer depths from Argo floats and two CMIP5-class climate models with a 1° spatial resolution, showing that shallow mixed layers are observed even during the winter in the Southern Ocean. These short-lived events could generate small phytoplankton blooms that keep the total biomass from decreasing to the

low concentrations seen in the models (Jönsson et al., 2015). The difference in Polar phytoplankton biomass between models and satellite-derived products is an area in need of more research.



# 5  Conclusions

In this study, we have shown that using probability distributions of Chl provides a comprehensive approach to compare biogeo-
chemical models with *in situ* data and satellite-derived fields. Direct point-by-point comparisons can be prone to overestimating
errors due to small temporal lags or displacements in space, while the ability for a model to generate a probability distribu-
tion function that matches well with the observed data suggests that physical and biological processes are resolved reasonably
well. We also found that Longhurst provinces act as a good classification system to use when generating the probability dis-
tributions, since they are defined to minimize within-region variability by separating areas that are controlled by different
physico-chemical processes from one another. Finally, EMDs provided a powerful approach to quantify the difference between
distributions in an objective way. The combined use of PDFs, Longhurst provinces, and EMDs allowed us to identify Longhurst
provinces such as the Polar oceans and tropical North Atlantic Ocean which need specific attention, and areas where the model
already shows a lot of skill. It is clear that model versus data comparisons and skill assessments need to be conducted in such
a way that one can start to address the specific processes and conditions that lead to discrepancies.

*Code and data availability.*  Example code for the processing and analysis can be found in the code repository https://doi.org/10.5281/
ZENODO.6683849 (Jönsson, 2022). Data used in the study can be accessed at https://doi.org/10.5285/D62F7F801CB54C749D20E736D4A1039F
(OC-CCI) and https://doi.org/DOI:10.7910/DVN/08OJUV (Darwin-CBIOMES-0).

*Author contributions.*  BJ initiated the project, performed most statistical analysis, and generated most figures. BJ also lead writing the text.
JB and SH lead the EMD analysis, SD, CF, GF, and CH lead the development of the Darwin output and provided expertise in how to interpret
the fields. TJ and SS lead the development of OC-CCI and provided expertise in how to interpret the product. The full authorgroup worked
closely together to develop the method and write the paper in a collaborative fashion.

*Competing interests.*  The authors declare no competing financial interests.

*Acknowledgements.*  This work was carried out under the auspice of the Simons Collaboration on Computational Biogeochemical Modeling
of Marine Ecosystems (CBIOMES), which seeks to develop and apply quantitative models of the structure and function of marine microbial
communities at seasonal and basin scales. Funding for this work was provided by the Simons Foundation (549947 SS, 553242, 549939, and
827829, C.L.F). Additional support from the National Centre for Earth Observations of the UK is acknowledged.



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





## List of Figures







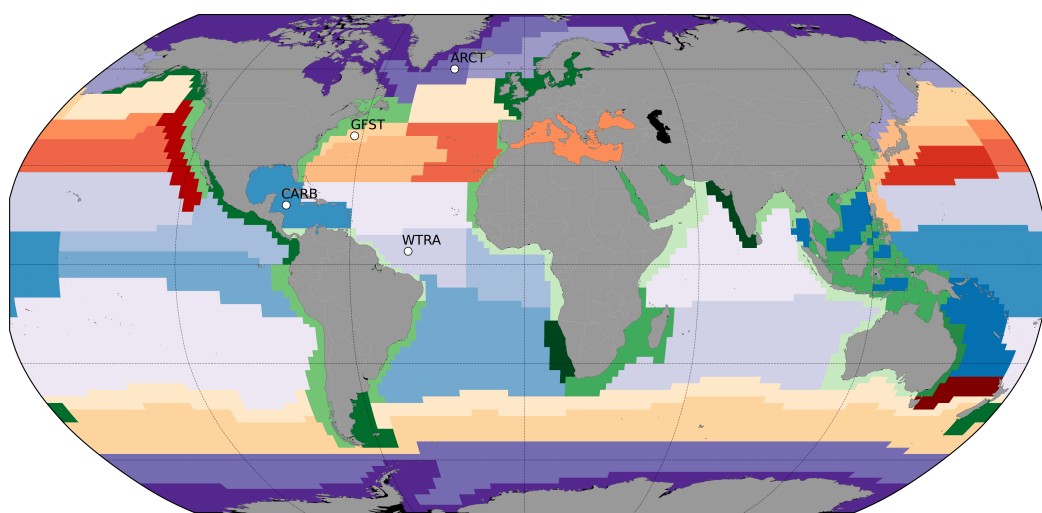

**Figure 1.** Biogeochemical provinces according to Longhurst (2007). Purple shades denote the Polar biome, red-yellow the Westerlies biome, blue the Trades biome, and green the Coastal biome. Regions analyzed in detail in this paper are identified by the codes ARCT (Atlantic Arctic Province), GFST (Gulf Stream Province), WTRA (Western Tropical Atlantic Province), and CARB (Caribbean Province).



**Figure 2.** Taylor diagrams based on comparisons between satellite-derived OC-CCI Chl ($Chl_{sat}$) and Chl concentrations from the the Darwin-CBIOMES-0 configuration ($Chl_{Rrs}$). Each point represents a Longhurst province. Model Standard Deviation is normalized to satellite data (thick dashed line). Different colors denote different basins: cyan = Arctic Ocean, red = Atlantic Ocean, blue = Indian Ocean, green = Pacific Ocean, black = Southern Ocean. Different marker shapes denote different biomes: triangle = Coastal, plus = Polar, circle = Trades, square = Westerlies. Panel A: Point-by-point comparison between $Chl_{sat}$ and $Chl_{mod}$ where daily match-ups and each grid cell in the model and satellite products are used. Panel B: the same as A but for $Chl_{sat}$ and $Chl_{Rrs}$. Panel C: Daily match-ups between $Chl_{sat}$ and $Chl_{mod}$, but all data falling within a Longhurst region is averaged to one value for each day for each Longhurst region. Panel D: the same as C but for $Chl_{sat}$ and $Chl_{Rrs}$. Panel E: The daily time series in panel B are further aggregated to monthly averages for matchups between between $Chl_{sat}$ and $Chl_{mod}$. Panel F: the same as E but for $Chl_{sat}$ and $Chl_{Rrs}$.





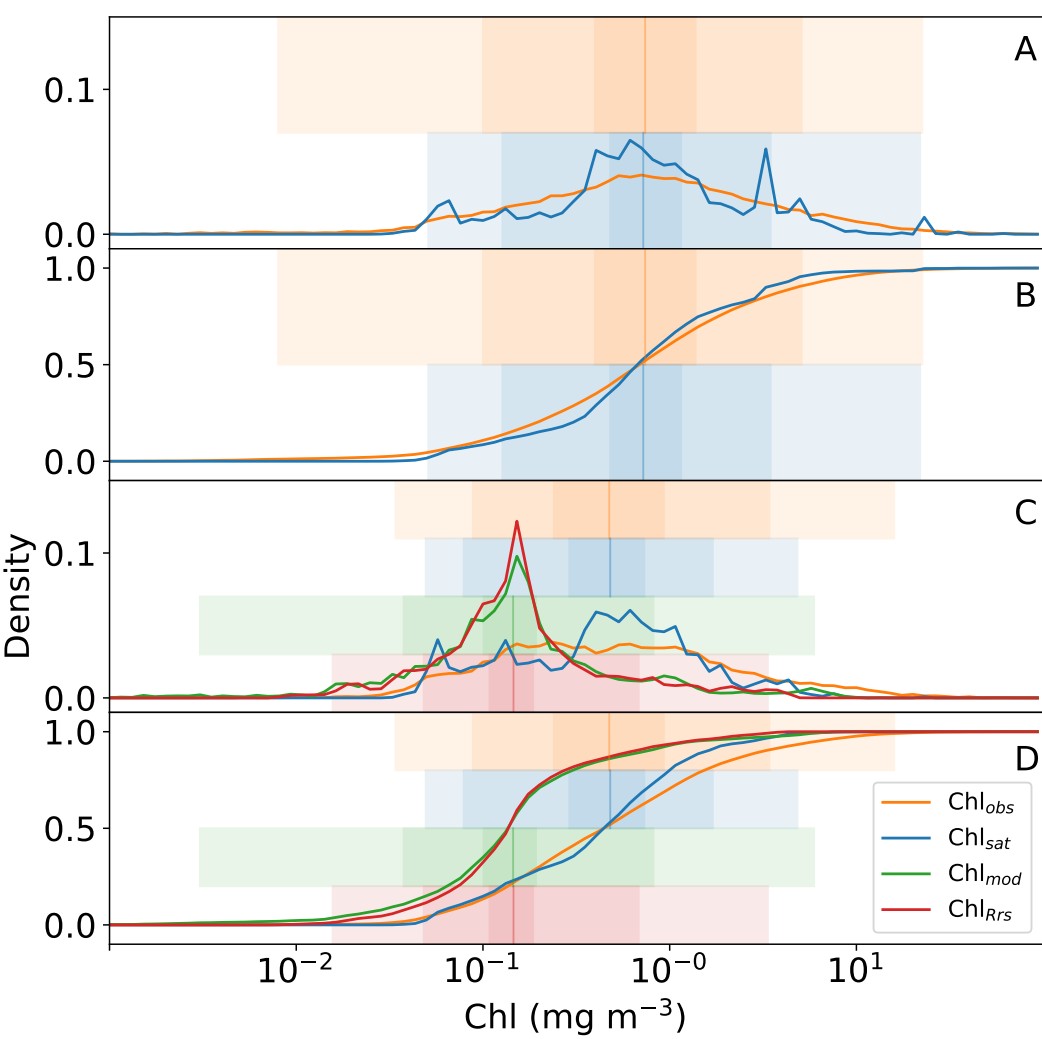

**Figure 3.** Distributions of *in situ* (Chl$_{obs}$), satellite derived (Chl$_{sat}$), modeled (Chl$_{mod}$) chlorophyll (Chl), and Chl derived from simulated Remote Sensing Reflectances in the Darwin-CBIOMES-0 configuration (Chl$_{Rrs}$). All datasets are matched in time and location and only complete match-ups are used. Shadings show the 1%–99%, 10%-90%, and ±1σ percentiles in the respective distributions. Panel A shows the respective histograms of Chl$_{obs}$ (orange line) and Chl$_{sat}$ (blue line). Panel B shows the corresponding cumulative distribution. Panels C and D are analogous to A-B, but with Chl$_{mod}$ (green lines) and Chl$_{Rrs}$ (red lines). Note that the datasets for Chl$_{obs}$ and Chl$_{sat}$ differ between panels A-B and C-D due to the smaller coverage in temporal range of Darwin-CBIOMES-0.



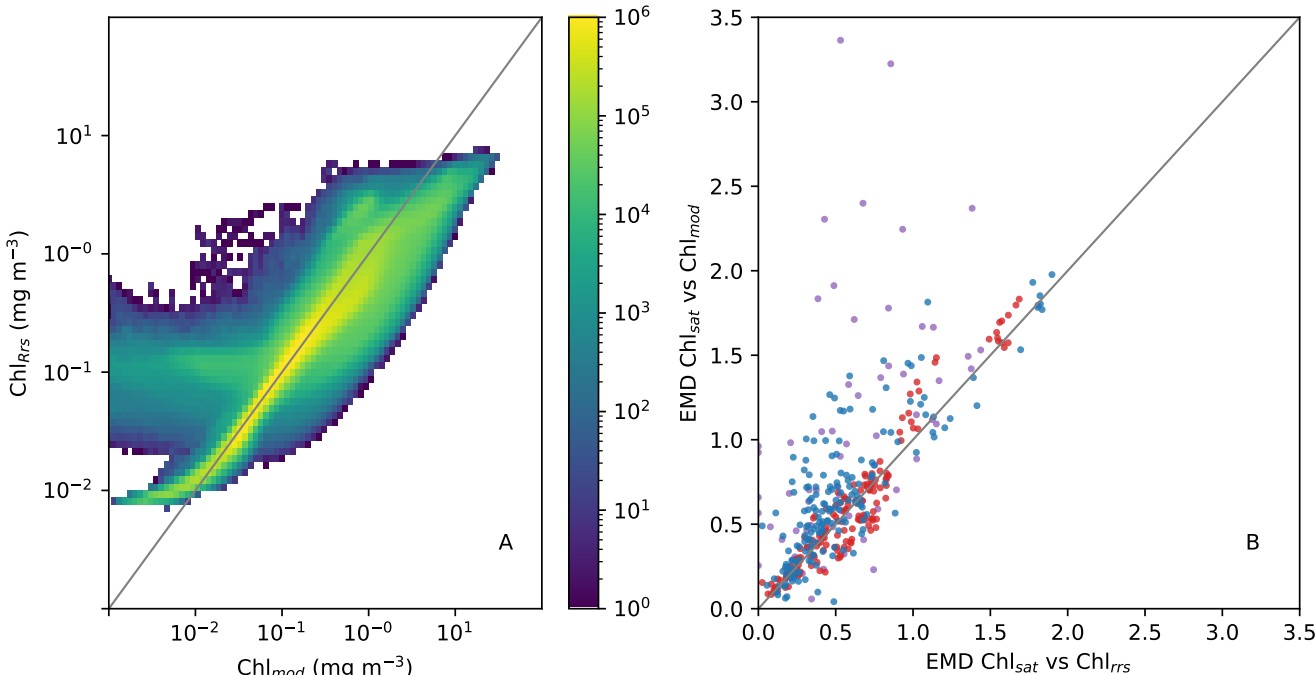

**Figure 4.** A) Point-by-point comparisons of satellite derived ($Chl_{sat}$), modeled ($Chl_{mod}$) chlorophyll. Colors denote number of data points in that bin. B) Earth Mover Distances for each province and month calculated using $Chl_{sat}$ and either $Chl_{mod}$ or Chl derived from simulated Remote Sensing Reflectances in the Darwin-CBIOMES-0 configuration ($Chl_{Rrs}$). Purple denotes the Polar biome, red the Westerlies biome, and blue the Trades biome. The Coastal biome is omitted as to which described in the main text.




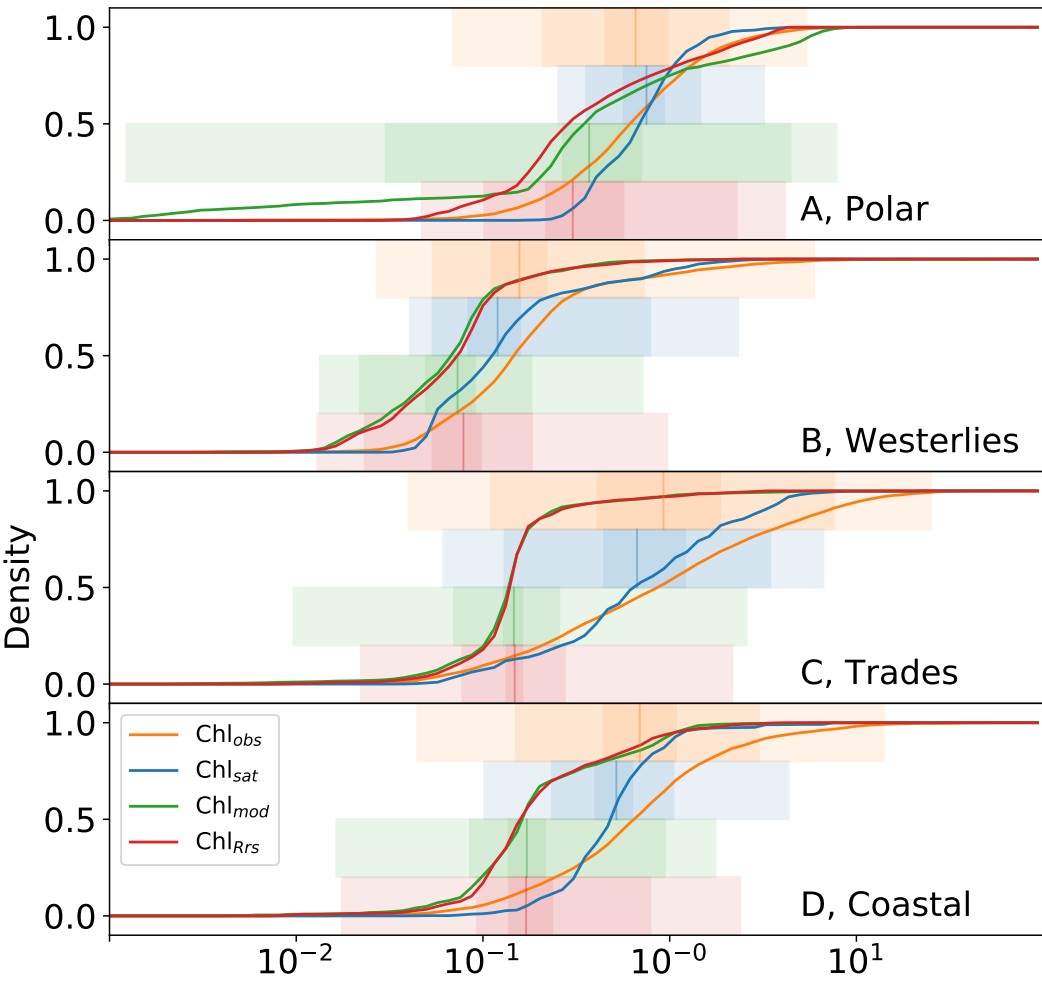

**Figure 5.** Distributions of *in situ* (Chl$_{obs}$), satellite derived (Chl$_{sat}$), modeled (Chl$_{mod}$) chlorophyll (Chl), and Chl derived from simulated Remote Sensing Reflectances in the Darwin-CBIOMES-0 configuration (Chl$_{Rrs}$) for each of Longhurst's biomes. All datasets are matched by time and location and only match-ups where data from all four sources are present are used. Shadings show the 1%–99%, 10%-90%, and ±1σ percentiles in the respective distributions. Panel A shows cumulative distributions for match-ups located in provinces in the Polar biome as defined by Longhurst (2007). Panel B shows analogous distributions for data in the Westerlies biome, panel C data for the Trades biome, and panel D data for provinces in the Coastal biome.



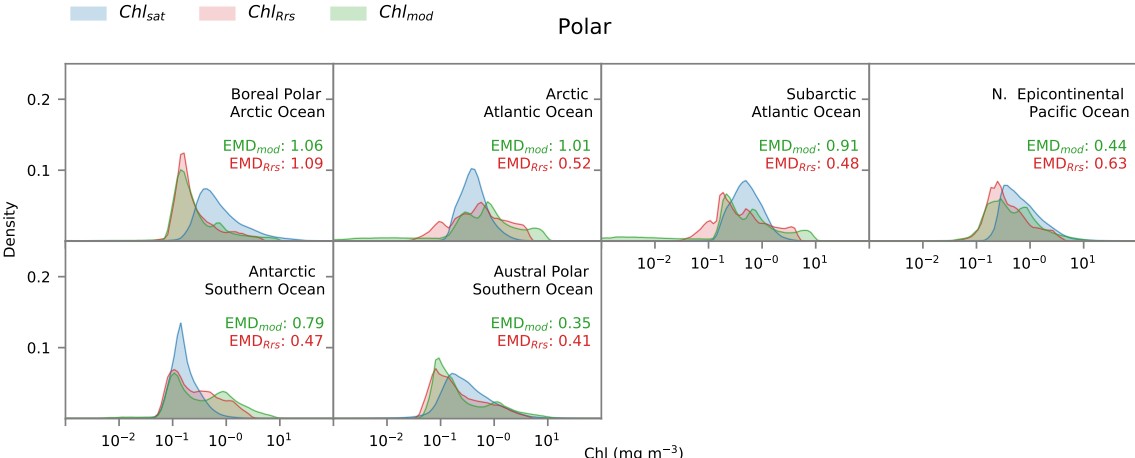

**Figure 6.** Distribution of satellite derived ($\text{Chl}_{sat}$, blue) and modeled ($\text{Chl}_{Rrs}$, red; $\text{Chl}_{mod}$, green) Chlorophyll for different Longhurst provinces in the Polar biome. Red values are Earth Mover's Distances (EMDs) between the $\text{Chl}_{sat}$ and $\text{Chl}_{Rrs}$ distributions, green values EMDs between $\text{Chl}_{sat}$ and $\text{Chl}_{mod}$ distributions.



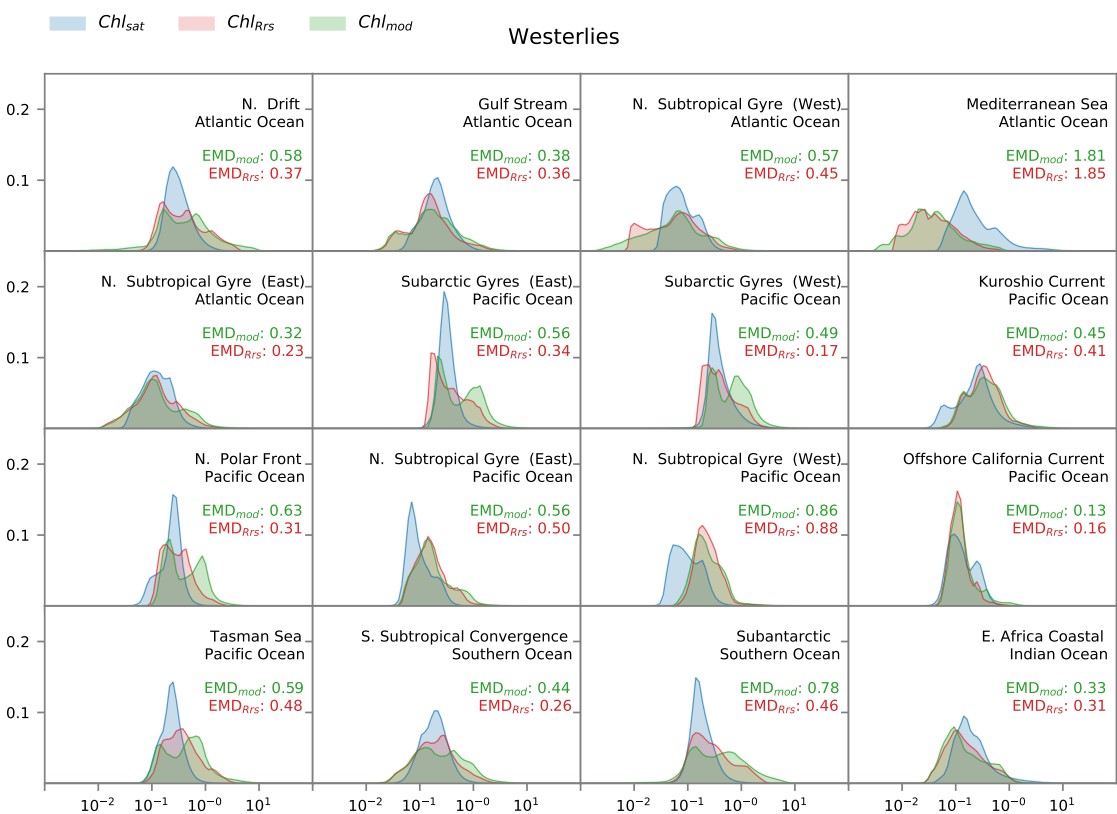

**Figure 7.** Distribution of satellite derived (Chl$_{sat}$, blue) and modeled (Chl$_{Rrs}$, red; Chl$_{mod}$, green) Chlorophyll for different Longhurst provinces in the westerlies biome. The East Africa Coastal Province is included as it contains the Aghulas current. Red values are Earth Mover's Distances (EMDs) between the Chl$_{sat}$ and Chl$_{Rrs}$ distributions, green values EMDs between Chl$_{sat}$ and Chl$_{mod}$ distributions.





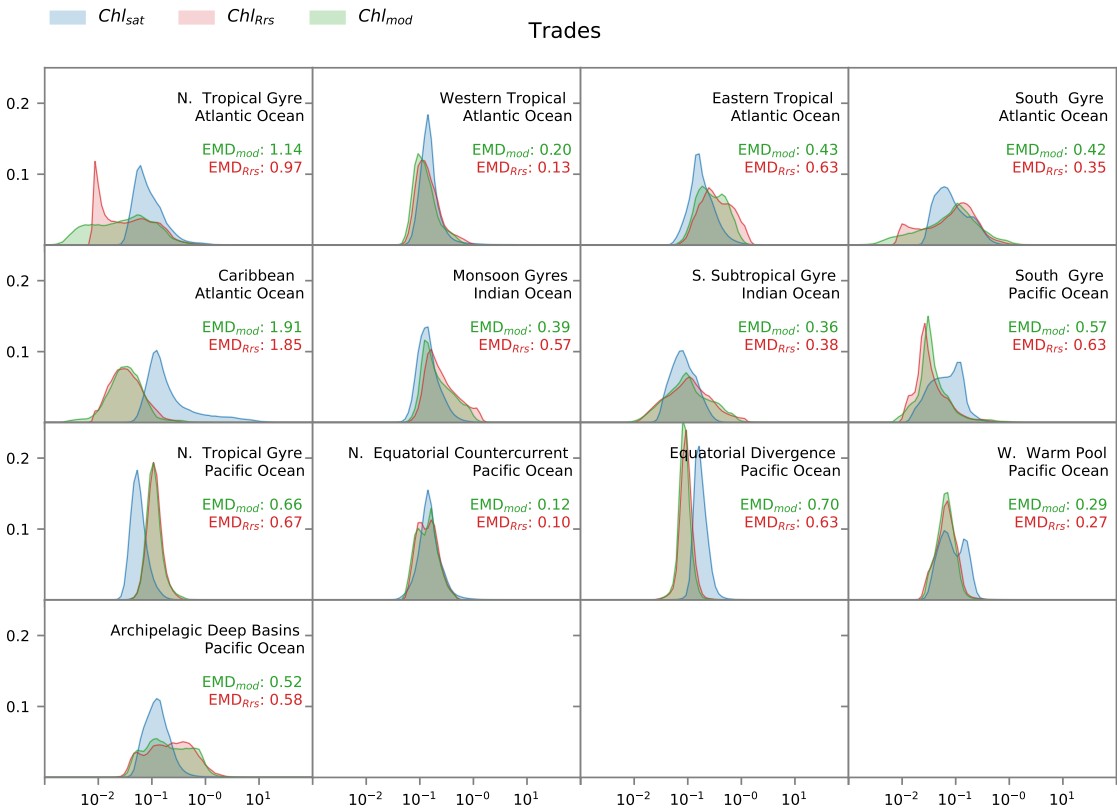

**Figure 8.** Distribution of satellite derived (Chl$_{sat}$, blue) and modeled (Chl$_{Rrs}$, red; Chl$_{mod}$, green) Chlorophyll for different Longhurst provinces in the trade winds biome. Red values are Earth Mover's Distances (EMDs) between the Chl$_{sat}$ and Chl$_{Rrs}$ distributions, green values EMDs between Chl$_{sat}$ and Chl$_{mod}$ distributions.




**Figure 9.** Monthly distributions of satellite derived (Chl$_{sat}$, blue), derived from model $R_{rs}$ (Chl$_{Rrs}$, red), and simulated Chl (Chl$_{mod}$, green). Shadings show the 1%–99%, 10%-90%, and ±1σ percentiles in respective distributions. Black vertical lines denote the medians. Panel A represent the Polar biome's *Atlantic Subarctic province*, panel B the Westerlies biome's *Gulf Stream province*, panel C the Trades biome's *Western Tropical Atlantic province*, and panel D the Trades biome's *Caribbean province*. Note that data are not available for January and December due to ice cover and adverse satellite viewing conditions in panel A. Red values are EMDs between the Chl$_{sat}$ and Chl$_{Rrs}$ distributions, green values EMDs between Chl$_{sat}$ and Chl$_{mod}$ distributions.



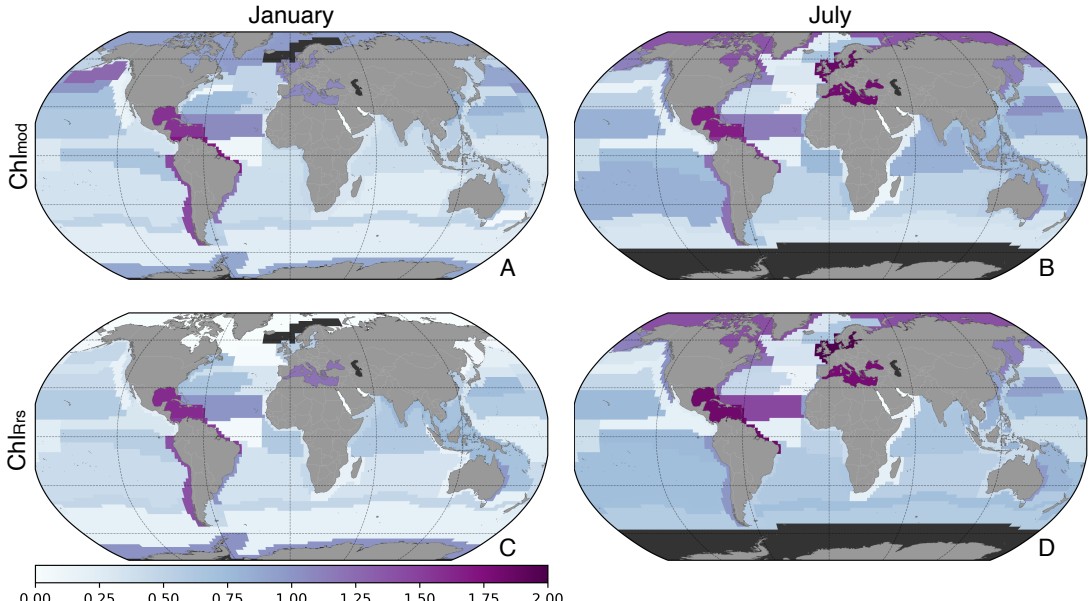

**Figure 10.** Map of Earth Mover Distances between $Chl_{sat}$ and $Chl_{mod}$ (panels A & B) or $Chl_{mod}$ (panels A & B) for different Longhurst provinces in January (panels A & C) or July (panels B & D). Dark grey color represents areas where data are not available.





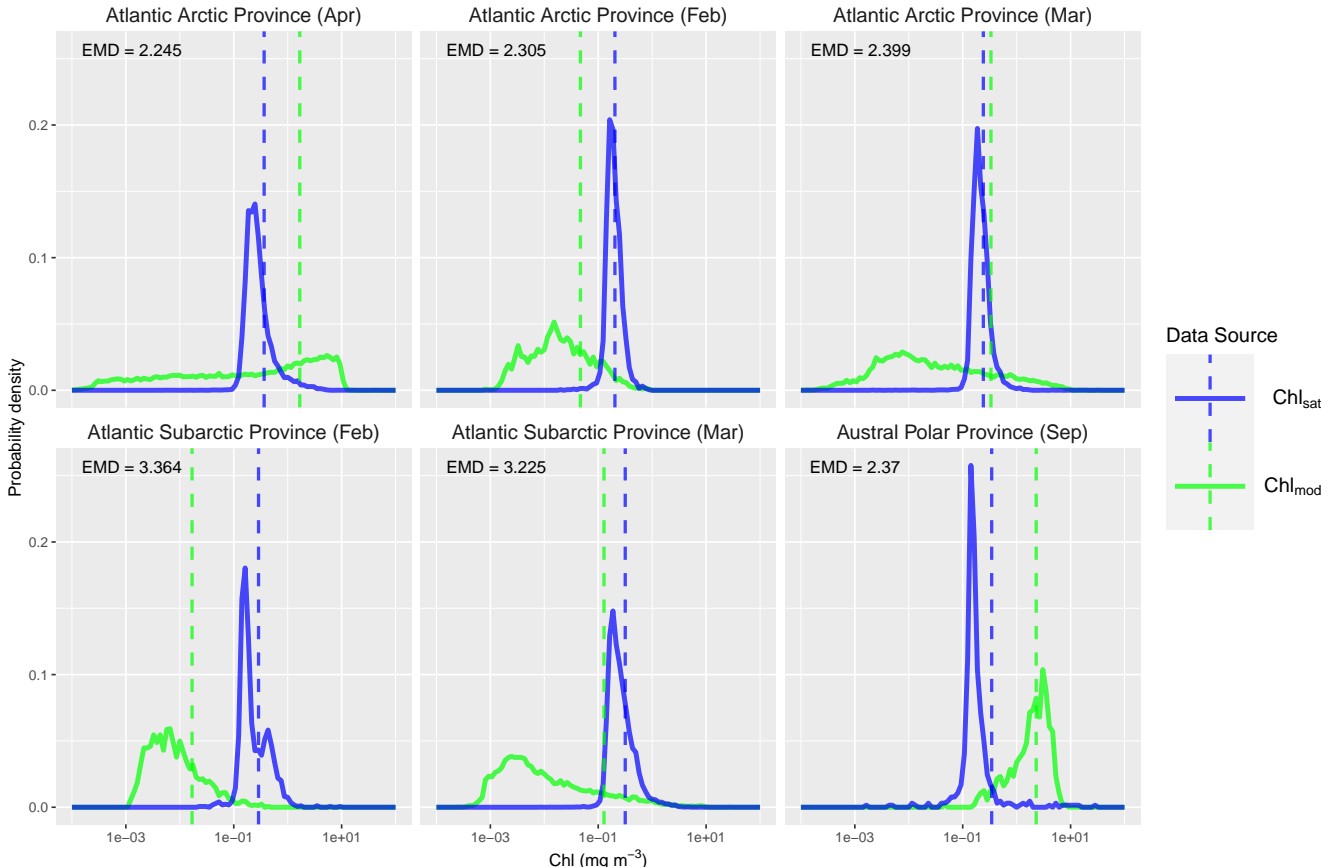

**Figure 11.** Distributions of satellite derived ($Chl_{sat}$, blue), and modeled Chl ($Chl_{mod}$, green) in the top six province-month pairs in terms of Earth Mover Distances (EMD). The vertical dashed lines mark the mean of each distribution. There appears to be a clear distributional difference between the two data sources in each case – a large amount of probability mass would have to be transported from one distribution to another to make the two equivalent. Also notable is that EMD can be large while the mean difference is small – this highlights how EMD is a richer measure of distributional difference.





**List of Tables**





**Table 1.** Earth Mover Distances comparing distributions of *in situ* ($Chl_{obs}$), satellite derived ($Chl_{sat}$), modeled ($Chl_{mod}$) chlorophyll (Chl), and Chl derived from simulated Remote Sensing Reflectances in the Darwin-CBIOMES-0 configuration ($Chl_{Rrs}$). See Figure 1 for the extent of each biome.

| Domain | $Chl_{obs}$ vs $Chl_{sat}$ | $Chl_{obs}$ vs $Chl_{Rrs}$ | $Chl_{obs}$ vs $Chl_{mod}$ |
|---|---|---|---|
| Global | 0.10 | 0.45 | 0.47 |
| Polar | 0.19 | 0.27 | 0.27 |
| Westerlies | 0.07 | 0.20 | 0.20 |
| Trades | 0.08 | 0.82 | 0.84 |
| Coastal | 0.35 | 0.57 | 0.58 |