# Peer review of "Using Probability Density Functions to Evaluate Models (PDFEM, v1.0) to compare a biogeochemical model with satellite derived chlorophyll"

_EGUsphere, 2022_

## Author Response (AR1)

**Using Probability Density Functions to Evaluate Models (PDFEM, v1.0) to compare a biogeochemical model with satellite derived chlorophyll**

Response to Reviewers

The authors would like to thank the reviewers for their insightful comments, which significantly helped improving the manuscript. Both reviewers feel that the manuscript makes a relevant contribution and should be published with minor modifications. Should the editor decide to accept our work, we have laid out the modifications that we would do before publication. They are all very straightforward. For each comment, we have first highlighted the issue, provided an answer, and described how the manuscript will be adjusted before publication. We are hopeful that the methods provided in our work will be of interest for the journal's audience and be helpful for many groups working with biogeochemical models.

**REVIEWER # 1**

*"The authors here present an interesting combination of statistical approaches to compare the probability density functions of observed and simulated chlorophyll across biomes and Longhurst provinces. I congratulate the authors on the work that has gone into this manuscript. It is extremely well written and the rationale behind their approach is cogently explained. I thoroughly hope that it encourages the ocean biogeochemical community to move beyond the use of correlation and RMSD as standard measures of model performance. I am happy to recommend the manuscript be accepted as it is but have a few very minor recommendations given below."*

**COMMENT # 1.1**

> Perhaps the authors could discuss how the approach might be extended to provide greater insight on model-observation mismatch? Where coincident physical observations are available, one could presumably subset model and observation data to access for example the ability of models to capture the impact of regional heat waves on chlorophyll distributions.

This is a good question and we are planning to at least add some more text about it in the discussion. Added text:

We believe that the skill of biogeochemical models to generate realistic distributions of properties are as, if not more, important than the skill to predict a property at a specific time an location or the long-term averages. Recent focus on regional heat waves (Oliver et al., 2021) and other extreme events have highlighted that rare physical conditions and consequent biological responses can have an outstanding influence on ocean health. It has also been suggested that the frequency of rare events might be as important as long-term averages to understand changes in marine ecosystems (Jönsson and Salisbury, 2016).

**COMMENT # 1.2**

> As most biogeochemical models do not produce a $Chl_{Rrs}$ output, perhaps the authors could offer some thoughts/recommendations on the use of the approach when only $Chl_{mod}$ output is available. The approach still seems to offer plenty of insight and this should perhaps be emphasized a bit more.

We will add some more recommendations and discussions about how to use $Chl_{mod}$. This is, however, a big topic that probably should be addressed systematically by the community. New text:

Our results show that comparisons between $Chl_{mod}$ and $Chl_{sat}$ are generally sufficient if $Chl_{Rrs}$ isn't available (which is normally the case) as long as these caveats are considered. Figures 6-9 can provide guidance to which regions where cautions should be applied.

**COMMENT # 1.3**

> L37 Should RMSD be standard deviation? While the RMSD is sometimes also given in Taylor plots this doesn't seem to be given in Figure 2.

This is a typo that we will fix. Thank you for finding it.

**Comment # 1.4**

> L80 Suggest giving the model name here.

We will add the model name as suggested.

**Comment # 1.5**

> Fig1. Suggest increasing the label font size for the regions identified.

We will increase the label font size.

**Comment # 1.6**

> Fig 2. A point color legend for these Taylor plots would be helpful.

It turns out that a legend becomes very unruly and hard to include.

**Comment # 1.7**

> Fig4a. The colorbar should have a label.

We will add a label to the colorbar.

**Comment # 1.8**

> Fig4b Suggest adding a point color legend.

We will add a color legend.

*I had the opportunity to review earlier versions of this work when submitted to another journal. I was in disagreement with the editorial decision because of the methodological merit of this work. I am very pleased to see the authors decided to rearranged the manuscript in the form of a model evaluation tool and presented in GMD. The last community effort on the validation of ocean biogeochemical models was published in a special issue of the Journal of Marine Systems in 2009 (some papers from that issue are referenced in the manuscript, like Jolliff et al. and Doney et al.). The authors have perfected the explanation, and the manuscript is very easy to read and to the point.*

*The concept of using PDFs to compare simulated properties is not new in the Earth Sciences and in climate sciences, but this is usually addressed through visual comparisons, or using changes in the descriptors (median, mean, skewness, etc). A recent effort from the SCOR WG on BGC model intercomparisons for simulations of the ocean iron cycle (FeMIP) advocated the use of PDFs to carry out this comparison (Rogerson and Vichi, 2021), but did not offer any objective way to measure the difference between the simulated and modelled distributions. This is exactly what the proposed method is about, and its application goes beyond the focus on chlorophyll, as clearly demonstrated by the authors with teh use of a variety of chl-related datasets (including model-derived,a s the irradiance based chl).*

*I would recommend swift publication with the very few technical corrections or suggestions listed below:*

COMMENT # 2.1

> L315: I have been impressed by how good the model is in the western boundary currents, and less in the eastern boundary currents. The good performance in the WBC is highlighetd in the discussion, but there is no discussion on the EBUS. I would expect the model to perform equally, given that in both cases primary production should occur at the surface. It may also be due to the broadness of EBUS, which are difficult to be separated from the adjacent sub-tropical gyres.

We have not discussed the EBUS regions in details as this is the topic of our next study, partly within the confines of the ESA funded project PRIMUS (`http://primus-atlantic.org`). We will add some more discussion about EBUSs in this paper as well. Added text:

Eastern Boundary Upwelling Systems show large discrepancies between $\text{Chl}_{\text{Rrs}}$ and $\text{Chl}_{\text{sat}}$. This is to be expected since these areas are characterized by complex interactions between physical and biological processes over short spatial scales. Other studies have also found that the dominating timescales of variability in these regions are very short, which most global biogeochemical models are not developed to resolve (Jönsson et al., 2023).

**COMMENT # 2.2**

L331-332: I think the author means the section on data availability. However, I could not find the figures in the Zenodo preview. I would suggest to upload all the supplementary figures to the repository, since some users may be interested in looking at the varous provinces without running the whole analysis

Great suggestion! We will add all figures to the github/Zenodo repository.

**COMMENT # 2.3**

L333-340 and Fig. 9: These are all example from the Northern Hemisphere. It would be interesting to see the seasonality in the Southern Ocean, especially because it is further addressed in the discussion. This is less necessary if all the supplementary figures are made available in the repository. Figure 9 can be graphically improved by adding all the labels of the months on the Y-axis, and also the tick labels in panel a and b. There is enough space to add the numbers there.

We will add all figures to the repository and include more labels.

**COMMENT # 2.4**

Sec. 3.6 I find this section title somehow confusing. The EMD has already been used previous results, (Fig.4b, 6, 7, and 8). Fig. 10 is a very useful spatial representation for determining the provinces with the higher discrepancy, so maybe the title should reflect the content of the section better. These graphs are however a little misleading, because the provinces that have a higher coastal coverage tend to have the higher EMD value. This is biased by the different distribution of lands between the northern and southern hemispheres. The Southern Ocean EMD is also quite high in July (as reported in the discussion), but this is somehow not coming out clearly.

Good point, we will change the section header to "Global and seasonal patterns of Earth Mover's Distances" and add more figures to the appendix.

COMMENT # 2.5

> Another point is that Fig. 11 and Fig. 10 are not showing the same months.
> From the text, one would expect to find in Fig. 11 the same months shown in
> Fig 10. Maybe some more explanation of the choice of the provinces in Fig. 11
> would help the reader.

The months presented in figure 10 are chosen to represent the Boreal and Austral
winter/summer (January and July). The panels in figure 11 show the Longhurst
regions and Months with highest with highest EMDs to present the variation in dis-
tributions. Most months in figure 11 are close to each other (Feb, Mar, Apr) and not
necessary the months that are globally most interesting. We will clarify the text.

COMMENT # 2.6

> L375-386 I agree entirely with the point made on the Polar Biomes, and this
> analysis is very clear. However, the consequence seems to be that models tend
> to have earlier blooms, rather than delays (Hague and Vichi, 2018). This is also
> related to the point made at lines 435-439. The suggested paper does support
> the same mechanism suggested by the authors.

We will rewrite the text to address this concern.

COMMENT # 2.7

> L387-389 The author have demonstrated the power of this method with the
> Darwin-CBIOMES model, but it can be used with any other biogeochemical
> model (restricting the analysis to $chl_{mod}$). I would suggest to make this point
> somewhere in the discussion

Please see the response to reviewer 1.